# Add-On Cyclic Angiotensin-(1-7) with Cyclophosphamide Arrests Progressive Kidney Disease in Rats with ANCA Associated Glomerulonephritis

**DOI:** 10.3390/cells11152434

**Published:** 2022-08-05

**Authors:** Domenico Cerullo, Daniela Rottoli, Daniela Corna, Mauro Abbate, Ariela Benigni, Giuseppe Remuzzi, Carlamaria Zoja

**Affiliations:** Istituto di Ricerche Farmacologiche Mario Negri IRCCS, Centro Anna Maria Astori, Science and Technology Park Kilometro Rosso, Via Stezzano 87, 24126 Bergamo, Italy

**Keywords:** crescentic glomerulonephritis, myeloperoxidase, anti-neutrophil cytoplasmic antibodies, cyclophosphamide, angiotensin-(1-7), glomerular crescents, inflammatory cells, parietal epithelial cells

## Abstract

Rapidly progressive crescentic glomerulonephritis associated with anti-neutrophil cytoplasmic antibodies (ANCA-GN) is a major cause of renal failure. Current immunosuppressive therapies are associated with severe side effects, intensifying the need for new therapeutic strategies. The activation of Mas receptor/Angiotensin-(1-7) axis exerted renoprotection in chronic kidney disease. Here, we investigated the effect of adding the lanthionine-stabilized cyclic form of angiotensin-1-7 [cAng-(1-7)] to cyclophosphamide in a rat model of ANCA-GN. At the onset of proteinuria, Wistar Kyoto rats with ANCA-GN received vehicle or a single bolus of cyclophosphamide, with or without daily cAng-(1-7). Treatment with cAng-(1-7) plus cyclophosphamide reduced proteinuria by 85% vs. vehicle, and by 60% vs. cyclophosphamide, and dramatically limited glomerular crescents to less than 10%. The addition of cAng-(1-7) to cyclophosphamide protected against glomerular inflammation and endothelial rarefaction and restored the normal distribution of parietal epithelial cells. Ultrastructural analysis revealed a preserved GBM, glomerular endothelium and podocyte structure, demonstrating that combination therapy provided an additional layer of renoprotection. This study demonstrates that adding cAng-(1-7) to a partially effective dose of cyclophosphamide arrests the progression of renal disease in rats with ANCA-GN, suggesting that cAng-(1-7) could be a novel clinical approach for sparing immunosuppressants.

## 1. Introduction

Pauci-immune crescentic glomerulonephritis is the most common form of rapidly progressive crescentic glomerulonephritis (RPGN), characterized by the lack of significant IgG deposits within glomeruli [1,2]. It is the renal manifestation of small vessel vasculitis associated with anti-neutrophil cytoplasmic antibodies (ANCA), mainly directed against proteinase 3 or myeloperoxidase (MPO) [1,3]. Clinically, RPGN causes a rapid loss of renal function and is often associated with hypertension, marked hematuria and massive proteinuria. Histologically, the typical disease lesion is the glomerular crescent, defined as two or more cell layers in the Bowman’s urinary space that progressively obliterate the glomerular tuft [2]. A mixture of resident parietal epithelial cells, podocytes and infiltrating monocytes/macrophages participate in the formation and progression of glomerular lesions [4,5].

ANCA-vasculitis is a life-threatening disease; its management consists of inducing remission with cyclophosphamide or rituximab, combined with glucocorticoids, followed by maintenance of remission to prevent relapse. However, this immunosuppressive approach is often associated with serious side effects, and 30–40% of patients may experience refractory disease or relapses [6], intensifying the need to find new therapeutic strategies that are more effective and less toxic.

Angiotensin-converting enzyme 2 (ACE2)/angiotensin 1-7 (Ang-1-7)/Mas receptor axis is the counter-regulatory pathway of the harmful action of angiotensin II (Ang II), which is one of the major contributors to the progression of chronic kidney disease [7,8]. ACE2 limits Ang II production through two mechanisms. On the one hand, it degrades Ang II to the vasodilatative heptapeptide Ang-(1-7); on the other hand, it converts Ang I to the inactive nonapeptide Ang-(1-9), which in turn can be converted to Ang-(1-7) by ACE [9]. Ang-(1-7) acts through the interaction with the G-protein-coupled receptor Mas [9,10]. Studies have shown that the administration of Ang-(1-7) had renoprotective effects in experimental models of kidney disease, including anti-Thy-1 glomerulonephritis [11], anti-GBM glomerulonephritis [12], type 2 diabetes [13] and Alport syndrome [14]. However, the short-half life in plasma of Ang-(1-7) because of a rapid in vivo catabolism by ACE and other proteases, is a limiting factor for its use in clinics [15]. Thioether cyclization can enhance the receptor specificity and stability of peptides [16,17]. An engineered cyclic (c)Ang-(1-7) peptide, obtained through a thioether-cyclization that confers enhanced proteolytic resistance and improved activity compared to the linear counterpart, has been developed [18]. We showed that treatment with the lanthionine-stabilized cAng-(1-7) offered renoprotection in BTBR *ob*/*ob* mice with type 2 diabetic nephropathy by limiting albuminuria, inflammatory cell infiltration, podocyte injury and glomerular and peritubular capillary rarefaction [19]. In addition, cAng-(1-7) has been reported to have a beneficial effect on glucose metabolism in type 1 and type 2 diabetes mouse models [20].

Here, we investigated the effect of treatment with cAng-(1-7) on top of cyclophosphamide in a rat model of MPO-ANCA-associated GN, which replicates the characteristic features of the disease in humans [21]. Cyclophosphamide was administered at a dose that, per se, induced only partial protection against renal disease.

## 2. Materials and Methods

### 2.1. Animals

Male Wistar Kyoto (WKY/NCrlBR) rats were purchased from Charles River Laboratories Italia (Calco, Lecco, Italy) and maintained in a specific pathogen-free facility with a 12 h dark/12 h light cycle, in a constant temperature room and with free access to a standard diet and water.

ANCA-associated crescentic glomerulonephritis was induced in WYK rats (180–220 g), as previously described [21]. Briefly, on day 0, rats were intramuscularly immunized with human myeloperoxidase (MPO, Elastin Products Company, Inc., Owensville, MO, USA) reconstituted in sterile water for injection at a concentration of 1000 µg/mL and suspended in an equal volume of complete Freund’s adjuvant (CFA, Sigma-Aldrich, St. Louis, MO, USA). The MPO-immunized rats also received 400 ng of pertussis toxin (Sigma-Aldrich) intraperitoneally (i.p) on days 0 and 2. On day 19, the disease was triggered by an intravenous (i.v.) injection of a sub-nephritogenic dose (0.02 mL/kg) of sheep anti-rat GBM serum (Probetex, Inc, San Antonio, TX, USA). On day 21, when animals had already developed hematuria and proteinuria, they were allocated randomly to receive (*n* = 5–6 rats/group) the following: vehicle (saline, by daily subcutaneous injection); the lanthipeptide cAng-(1-7) (1 µg/kg/day by subcutaneous daily injection; Lanthio Pharma, Groningen, The Netherlands) plus cyclophosphamide (single bolus 50 mg/kg i.p.); or cyclophosphamide (50 mg/kg single bolus 50 mg/kg i.p.). Cyclophosphamide was administered in a single bolus on day 21. A group of unimmunized WKY rats (*n* = 5) served as controls. All animals were followed until day 35, when they were euthanized by CO_2_ inhalation.

### 2.2. Laboratory Parameters

Twenty-four-hour urine samples were collected using metabolic cages on days 21, 28, 35 post-immunization. Hematuria was assessed using dipstick Multistix (Siemens, Munich, Germany). Urinary protein excretion was measured using the Coomassie method with a Miura one auto-analyzer (I.S.E. S.r.l. Rome, Italy). Renal function was assessed as blood urea nitrogen (BUN) in whole blood samples using the Reflotron test (Roche Diagnostic Corporation, Indianapolis, IN, USA). Systolic blood pressure was measured with a computerized tail-cuff system in conscious rats (BP-2000 Blood Pressure Analysis System, Visitech System; Apex, White Oak, NC, USA).

### 2.3. Histological Analysis

Duboscq-Brazil-fixed, paraffin-embedded renal sections (3 μm) were stained with periodic acid-Schiff reagent (PAS) (Bio-Optica, Milan, Italy), and Masson’s trichrome staining to analyze glomerular and tubular interstitial injury. The presence of at least two layers of proliferating cells in the Bowman’s space of the glomerular cross-sectional area was defined as a crescentic glomerular lesion. The mean percentage of glomeruli with crescents was estimated through the analysis of all glomeruli for each section. Tubular damage (atrophy and dilatation) was graded from 0 to 4+ (0, no changes; 1, changes affecting <25% of the sample; 2, changes affecting >25–50% of the sample; 3, changes affecting >50–75% of the sample; 4, changes affecting >75–100% of the sample). The number of tubular casts was counted in an average of 15 fields of interstitial areas (HPF/×200). All biopsy specimens were examined by two pathologists blind to treatment groups.

### 2.4. Immunofluorescence Analysis

For immunofluorescence experiments, OCT-frozen kidney sections (3 μm thick) were fixed with cold acetone and incubated with 1% BSA to block nonspecific sites. For the quantification of neutrophils, CD4+ T cells and endothelial rarefaction, the following primary antibodies were used: mouse anti-rat granulocytes (HIS48) (1:100; ab33760 Abcam, Cambridge, UK), mouse monoclonal anti-rat CD4 (1:25; MCA55G Bio-Rad, Hercules, CA), mouse monoclonal anti-rat RECA-1 (1:100; MCA970R Bio-Rad, Hercules, CA, USA) followed by the specific Cy3-conjugated secondary antibodies. Claudin-1/NCAM double stainings were performed through incubation with rabbit anti-claudin-1 (undiluted; Thermo Scientific, Rockford, IL, USA), mouse monoclonal anti-rat NCAM (1:4; Developmental Studies Hybridoma Bank, University of Iowa, Iowa City, IA, USA) followed by the specific FITC- or Cy3-conjugated secondary antibodies. Nuclei were stained with DAPI, and the renal structure with FITC-wheat germ agglutinin (WGA). Negative controls were obtained by omitting primary antibodies on adjacent sections. For the detection of claudin-1 and NCAM stainings, antigen retrieval with citrate buffer was performed to enhance the reactivity of antibodies to antigens. Fluorescence was examined using an inverted confocal laser-scanning microscope (Leica TCS SP8, Leica Microsystems, Wetzlar, Germany). At least 30 glomeruli were examined for each animal, and the extent of altered distribution of claudin-1 in the glomeruli was expressed as a score from 0 to 3 related to the percentage of the glomerular tuft area occupied by the claudin-1-positive cells (0: claudin-1 constitutively expressed by all PECs lining the Bowman’s capsule, 1: altered distribution of claudin-1 affecting ≤25% of the tuft, 2: altered distribution of claudin-1 affecting 25% to 50% of the tuft, 3: altered distribution of claudin-1 affecting >50% of the tuft). HIS48-positive neutrophils in the glomeruli, and CD4+ cells within glomeruli, were counted in an average of 25 glomeruli in each kidney sample and expressed as the average number of cells per glomerulus. RECA-1 staining of capillaries in the glomerular tuft was scored as previously described [22] as follows: 0, normal density of capillaries; 1+, reduced capillary density in <25% of tuft; 2+, reduced capillary density in 25–50% of the tuft; and 3+, reduced capillary density in >50% of the tuft.

### 2.5. Immunoperoxidase Analysis

ED-1 and CD8 stainings were performed using the immunoperoxidase method. Duboscq-Brazil-fixed, 3 μm paraffin-embedded kidney sections were incubated for 5 min with Peroxidazed 1 (PX968, Biocare Medical, Pacheco, CA, USA) to quench endogenous peroxidase, after antigen retrieval in Rodent Decloaker (RD913, Biocare Medical). After blocking for 10 min with Rodent Block R (RBR962, Biocare Medical), sections were incubated with mouse monoclonal antibody to monocyte/macrophage ED-1 surface antigen (1:100; MAB1435 Merck Millipore, Burlington, MA, USA) or mouse anti-rat-CD8 (1:50; 550298 DB Biosciences) followed by Rat HPR-Polymer (BRR4016, Biocare Medical) for 30 min. The stainings were visualized by the addition of the betazoid 3,3′diaminobenzidine chromogen kit solutions (BDB2004H, Biocare Medical). Finally, slides were counterstained with Mayer hematoxylin and observed through light microscopy (ApoTome, Axio Imager Z2, Zeiss, Oberkochen, Germany). Negative controls were obtained by omitting the primary antibody on adjacent sections. ED-1-positive monocytes/macrophages and CD8+ T cells within glomeruli were counted in an average of 25 glomeruli and expressed as the average number of cells per glomerulus.

### 2.6. Ultrastructural Analysis

Fragments of kidney tissue were fixed overnight in 2.5% glutaraldehyde (Sigma-Aldrich, St. Louis, MO) in 0.1 M sodium cacodylate buffer (pH 7.4) (Electron Microscopy Sciences, Hatfield, PA, USA) and washed repeatedly in the same buffer. After postfixation in 1% OsO4, specimens were dehydrated through ascending grades of alcohol and embedded in Epon resin. Ultrathin sections were stained with Uranyless (Electron Microscopy Sciences) and lead citrate (Electron Microscopy Sciences) and examined using a transmission electron microscopy (Morgagni 268D, Philips, Brno, Czech Republic).

### 2.7. Statistical Analysis

Data are presented as the mean ± SEM. Data analysis was performed using Prism Software (GraphPad Software Inc., San Diego, CA, USA). Comparisons were made using ANOVA with the Tukey post hoc test or two-way ANOVA for repeated measures. Statistical significance was defined as *p* < 0.05.

## 3. Results

### 3.1. Systemic Parameters

Data on the systemic parameters evaluated at the end of the study (day 35) are reported in Table 1. Rats with ANCA-GN receiving vehicle exhibited a significant reduction in body weight compared to healthy control rats. Both the treatment with cAng-(1-7), given daily for two weeks plus cyclophosphamide given as single bolus, and the treatment with cyclophosphamide alone, limited the loss of body weight compared to rats receiving vehicle.

ANCA-GN was associated with the development of systemic hypertension, which was reduced significantly by treatment with cAng-(1-7) plus cyclophosphamide, but not with cyclophosphamide alone.

Rats with ANCA-GN given vehicle exhibited kidney hypertrophy. A decrease in the kidney weight/body weight ratio was observed in both the combination therapy and the cyclophosphamide group, compared to the vehicle group. Notably, the kidney weights of rats treated with cAng-(1-7) plus cyclophosphamide were similar to those of control rats.

### 3.2. Addition of cAng-(1-7) to Cyclophosphamide Arrests Progression of Kidney Disease in Rats with ANCA-GN

Twenty-one days after immunization, when they were assigned to the experimental groups, WKY rats exhibited urine abnormalities for proteinuria and hematuria. In the vehicle group, urinary protein excretion progressively and significantly increased during the study (Figure 1A). Treatment with the combination of cAng-(1-7) and cyclophosphamide arrested the progression of proteinuria, stabilizing its levels and resulting in a significant decrease compared to the vehicle group from day 28. At the end of the study (day 35), proteinuria levels were similar to those measured at day 21 (pre-treatment). At day 28, cyclophosphamide alone also resulted in a significant reduction in urinary protein excretion. However, at variance with the combination therapy, in this group, at the end of the study (day 35), a rise in proteinuria was observed, with levels that were still lower compared to vehicle, but higher than those of combination therapy (*p* < 0.05). Indeed, the combined therapy afforded for a 60% reduction in proteinuria levels with respect to cyclophosphamide alone (Figure 1A). Hematuria was not affected by treatments (score 4 in all groups of rats with ANCA-GN vs. 0 in non-immunized rats).

According to our previous study [21], rats with ANCA-GN given vehicle exhibited renal function impairment, as revealed by higher levels of BUN compared to controls (Figure 1B). Both treatment with cAng-(1-7) plus cyclophosphamide and treatment with cyclophosphamide alone restored BUN to control levels.

### 3.3. Addition of cAng-(1-7) to Cyclophosphamide Ameliorates Renal Structure in Rats with ANCA-GN

WKY rats with ANCA-GN given vehicle exhibited the typical features of human disease, consisting of glomerular crescents at different stages, and tubulo-interstitial injury. Glomerular lesions (Figure 2A) involved 41 ± 2% of glomeruli, with a prevalence of fibrocellular crescents (Figure 2B,C). In addition to glomerular crescents, in 52 ± 2% of glomeruli, adhesions between the glomerular tuft and the Bowman’s capsule, and focal areas of necrosis were present, while only an average of 7 ± 2% of glomeruli were normal. Fibrinoid necrosis was evident in an average of 50 ± 7% glomeruli with crescents (Figure 2B). The administration of cAng-(1-7) on top of cyclophosphamide remarkably preserved kidney structure, limiting crescents to only 9 ± 1% of the glomeruli (*p* < 0.0001 vs. vehicle) (Figure 2B). Importantly, the analysis of the crescent composition of these few affected glomeruli revealed, unlike in the vehicle group, the prevalence of cellular crescents (Figure 2C). The percentage of normally appearing glomeruli, which had risen to 41 ± 5%, confirmed that combination therapy has a strong renoprotective effect (Figure 2B). ANCA-GN rats that received the single bolus of cyclophosphamide had 29 ± 3% of glomeruli affected by crescents, a percentage which was significantly (*p* < 0.05) lower compared to the vehicle group, but higher than the combination therapy group (*p* < 0.001) (Figure 2B). These crescents were predominantly fibrocellular, as was the case in the vehicle group, and were accompanied by fibrinoid necrosis in an average of 67 ± 4% crescents (*p* < 0.05 vs. cAng-(1-7) + cyclophosphamide) (Figure 2B,C).

### 3.4. Addition of cAng-(1-7) to Cyclophosphamide Restores PEC/Progenitor Cell Distribution

According to our previous observations [21], the glomerular crescents in rats with ANCA-GN were associated with a dysregulated pattern of parietal epithelial cells (PECs) and progenitor cells. In particular, we identified PECs through the specific marker claudin-1 and progenitor cells using neural cell adhesion molecule (NCAM). As shown in Figure 3A, glomeruli from control rats exhibited the typical linear pattern of claudin-1+ and NCAM+ cells along the Bowman’s capsule. In contrast, PECs with a progenitor phenotype were found in the glomerular tuft of rats that received vehicle. Treatment with cAng-(1-7) on top of cyclophosphamide re-established the normal distribution of PECs/progenitors in the majority of glomeruli, limiting the accumulation of these cells in the glomerular tuft. Treatment with cyclophosphamide alone only partially limited the alterations of PECs, which were abnormally displaced in a large part of glomeruli. The results of the quantification of the altered distribution of claudin-1-positive PECs in ANCA-GN rats and the effects of the treatments are shown in Figure 3B.

### 3.5. Addition of cAng-(1-7) to Cyclophosphamide Limits Glomerular Inflammation and Endothelial Rarefaction in Rats with ANCA-GN

In WKY rats with ANCA-GN, glomerular lesions were associated with a marked accumulation of neutrophils, monocytes/macrophages, CD4^+^ T cells and CD8^+^ T cells in the glomerular tuft (Figure 4A). Treatment with cAng-(1-7) on top of cyclophosphamide exerted an important anti-inflammatory effect by limiting glomerular accumulation of the different subtypes of cells. Specifically, combination therapy halved the number of glomerular cells that were positive for the granulocyte marker HIS48, compared to the vehicle group (Figure 4A). The staining for the monocyte/macrophage marker ED1 was markedly reduced in animals treated with cAng-(1-7) plus cyclophosphamide (Figure 4A). Glomerular accumulation of CD4^+^ T cells was prevented by the combination of cAng-(1-7) and cyclophosphamide, while CD8^+^ T cells were numerically limited compared to the vehicle group (Figure 4A). In contrast to the anti-inflammatory effect achieved by treatment with cAng-(1-7) plus cyclophosphamide, the single bolus of cyclophosphamide alone did not limit but actually worsened glomerular accumulation of inflammatory cells compared to the vehicle group (Figure 4A).

Next, we assessed whether, in rats with MPO-ANCA-GN, the anti-inflammatory effect of cAng-(1-7) plus cyclophosphamide was associated with an amelioration of glomerular endothelial damage. Figure 4B shows endothelial rarefaction, assessed through the endothelial marker (RECA-1), in the glomeruli of rats given vehicle. This alteration was evident mainly in the areas affected by crescents and their neighboring glomerular capillaries. Treatment with cAng-(1-7) plus cyclophosphamide, but not cyclophosphamide alone, reduced the area of the tuft affected by endothelial injury (Figure 4B).

### 3.6. Addition of cAng-(1-7) to Cyclophosphamide Ameliorates Ultrastructural Abnormalities in Glomeruli in Rats with ANCA-GN

Electron microscopy analysis of glomeruli from rats with ANCA-GN that were on vehicle showed crescentic lesions closely resembling human pathology, as previously described [21]. Fibrin deposits were associated with the accumulation of polymorphonuclear neutrophils, mononuclear cells, red blood cells, the disorganization and rupture of the GBM and of the Bowman’s capsule, and prominent glomerular epithelial cells in crescents (Figure 5A). Some areas in the adjacent tuft also exhibited segmental intracapillary fibrin and inflammatory cells with endothelial cytoplasmic enlargement or loss, and severe effacement of foot processes and vacuolization of podocytes (Figure 5A). The treatment with cAng-(1-7) plus cyclophosphamide caused the most dramatic protective effects (Figure 5B). Almost normal-looking glomeruli were prominent, and no ruptures were detected in the GBM or Bowman’s capsules, indicating the most pronounced focality and resolution of lesions. Inflammatory cells and capillary wall changes were observed in some areas (Figure 5B). Notably, unlike the abundant inflammatory cells detected in the glomeruli of the rats with ANCA-GN on vehicle (Figure 5A), many leukocytes appeared as circulating neutrophils with granular structures in their cytoplasm, without evidence of infiltration into capillary structures (Figure 5B). Similarly, glomerular endothelial abnormalities were mild and the effacement of the foot processes of podocytes was very focal (Figure 5B). The combined treatment markedly attenuated the collapse of capillary loops and signs of more advanced injury. In rats treated with cyclophosphamide alone, compared with the vehicle group, crescentic lesions were less frequent, in line with the light microscopy findings. The glomeruli exhibited variable degrees of acute damage (cell infiltration, capillary wall damage) or post-inflammatory adhesions of the glomerular tuft to the capsule and matrix accumulation.

### 3.7. Addition of cAng-(1-7) to Cyclophosphamide Ameliorates Tubulo-Interstitial Abnormalities in Rats with ANCA-GN

Light microscopy analysis of the tubulo-interstitial compartment of rats with ANCA-GN given vehicle showed tubular dilatation and atrophy (score: 1.08 ± 0.20 and 0.67 ± 0.11, respectively), accompanied by tubular casts (number of casts/field: 3.75 ± 0.71, 200×) (Figure 6A). In the group of rats that received cAng-(1-7) plus cyclophosphamide, a significant (*p* < 0.05) reduction in the number of proteinaceous casts (1.74 ± 0.16) was observed compared to vehicle; tubular dilatation (score: 0.60 ± 0.19) and atrophy (score: 0.70 ± 0.12) were no different from vehicle. In animals treated with a single bolus of cyclophosphamide, the number of proteinaceous casts averaged 2.37 ± 0.62 (NS vs. vehicle). Next, given the striking effect of the combined treatment on the glomerular capillary wall, we investigated whether a protective effect could be exerted on ultrastructural changes in post-glomerular vessels. The analysis of interstitial areas in rats with ANCA-GN on vehicle revealed swelling and loss of fenestrations of endothelial cells of peritubular capillaries associated with the accumulation of inflammatory cells, mainly monocytes/macrophages and lymphocytes (Figure 6B). In the kidneys of rats with cAng-(1-7) plus cyclophosphamide, peritubular capillary endothelial changes were clearly much more focal and milder. Inflammatory cells were absent or scattered and sparsely distributed (Figure 6B).

## 4. Discussion

In this study, we provide experimental evidence that daily treatment with the cyclic form of Ang-(1-7) on top of a single injection of cyclophosphamide has a deep renoprotective effect on rats with MPO-ANCA-associated crescentic GN. This therapeutic protocol blocked the rise of proteinuria and ameliorated renal function, dramatically reducing the number of glomeruli affected by crescents and limiting glomerular inflammation. Conversely, the administration of cyclophosphamide alone had partial effects on the course of the disease and pathology.

Here, the first remarkable finding is that combined therapy had a stable and pronounced effect on proteinuria, while evidence of a further rise in proteinuria at the end of the study in rats treated with cyclophosphamide alone indicates a tendency for the disease to relapse in the absence of co-administered cAng-(1-7). This may reflect the clinical condition of patients with ANCA-GN who do not completely respond to cyclophosphamide. Interestingly, the antiproteinuric effect of the combined treatment may be reminiscent of the data collected by our group from BTBR *ob*/*ob* mice with diabetic nephropathy, in which the addition of cAng-(1-7) to ACE inhibitor therapy had a superior effect to ACE inhibitor alone, thanks to better preservation of podocytes and an improvement in glomerular endothelial damage [19]. Likewise, cAng-(1-7) on top of cyclophosphamide preserved podocyte structure and mostly abolished endothelial rarefaction.

The improvement in proteinuria after treating ANCA-GN rats with c-(Ang-1-7) plus cyclophosphamide was associated with a very strong reduction in the number of glomeruli affected by crescents and an increased number of normal glomeruli. The trigger event in crescent formation in ANCA-GN is the development of necrotizing and inflammatory injury to the glomerular capillary wall, leading to gaps in the GBM, extravasation of plasma proteins and red cells, and a severe inflammatory reaction [23,24,25].

Studies reported a pivotal and interconnected role for neutrophils and monocytes/macrophages in the acute phase of the disease [26,27]. ANCA-activated neutrophils cause vascular injury through the release of radical oxygen species and lytic enzymes [28,29,30], and macrophages that accumulate in focal areas of the glomerular tuft along the GBM by releasing matrix metalloproteinase-12 cause the rupture of the GBM [23,27,31]. Serine proteases such as catepsin G, proteinase-3 and elastase, released by activated neutrophils, are also able to induce a local activation of the renin angiotensin system by cleaving angiotensinogen to Ang I and further to Ang II at the site of injury and inflammation [32,33]. Ang II is a major player in glomerular disease that, acting through the angiotensin II type 1 receptor, promotes vasoconstriction and capillary hypertension, drives extracellular matrix and fibrosis, and fuels glomerular inflammation [34]. In the present study, the addition of cAng-(1-7) to cyclophosphamide resulted in reduced glomerular accumulation of neutrophils and macrophages and preserved GBM structure. One reasonable explanation for these findings is based on the anti-inflammatory effect exerted by Ang-(1-7) via the MasR that can modulate neutrophil recruitment and macrophage accumulation [12,35,36,37]. In this regard, Ang-(1-7) has been reported to be capable of promoting the resolution of the inflammation by reducing the influx of neutrophils, inducing their apoptosis, and limiting cytokine production [36,38]. Notably, we found that a single administration of cyclophosphamide alone was not sufficient to hamper glomerular infiltration of neutrophils, possibly suggesting that the lack of cAng-(1-7) had an important effect on the course of the disease, leading to an acute phase of disease relapse as also indicated by the rise of proteinuria.

In the glomeruli of animals treated with combination therapy, we also found a reduced number of CD4+ T cells, which may be a consequence of the reduced number of neutrophils recruited to the glomerular tuft in response to therapy. Indeed, MPO released by activated neutrophils is known to act as planted antigen for anti-MPO effector CD4+ T cells [39] and, in this regard, we previously observed in our model the presence of glomerular extracellular deposition of MPO, associated with neutrophil influx [21]. Notably, in the group of rats that received the combined therapy, at variance with rats given vehicle, we observed through electron microscopy that the polymorphonuclear cells that were present in the glomerular capillary had a preserved structure, indicating that they did not undergo degranulation.

Ultrastructural analysis also revealed that, according to our previous study [21], rats with ANCA-GN given vehicle exhibited obliteration of glomerular capillary lumens by neutrophils and monocytes/macrophages associated with GBM disorganization and gaps. The lower rate of macrophages in the glomeruli of animals treated with cAng-(1-7) plus cyclophosphamide may well account for the preserved GBM found in this group. The extravasation of inflammatory cells through the interrupted GBM contributes to the formation of crescentic lesions that are composed of a mixture of cell types, including monocytes/macrophages, podocytes, and PECs [4,5,25]. It is noteworthy that macrophages can release Ang II, which, through the activation of Ang type 1 receptor, induces podocytes to release the stromal cell-derived factor-1, which binds to CXCR4 and CXCR71 expressed on PECs, promoting their activation and migration toward the tuft, thereby obliterating the urinary space [5]. Thus, in our experimental setting, the anti-inflammatory effect exerted by cAng-(1-7) added to cyclophosphamide, limited macrophages in the glomeruli, resulting in a preserved structure of the GBM and a restored pattern of progenitor PECs along the Bowman’s capsule. Notably, both PECs and macrophages contribute to the progression of crescentic lesions toward a more fibrotic and irreversible state [24,25,40]. According to this evidence, in the group of rats treated with cAng-(1-7) and cyclophosphamide, the limited PEC migration and macrophage accumulation were associated with the prevalence of cellular crescents, which are considered reversible. Finally, the anti-inflammatory effect of the treatment with cAng-(1-7) on top of cyclophosphamide, also involved the tubular compartment, where signs of capillaritis were almost abolished and associated with a preserved endothelium.

In conclusion, the addition of cAng-(1-7) to cyclophosphamide enhanced the renoprotective effect of cyclophosphamide alone in an experimental model of ANCA-GN. The combination therapy, but not cyclophosphamide alone, interrupted the vicious cycle of inflammation initiated by neutrophil recruitment and degranulation, resulting in reduced glomerular accumulation of macrophages and CD4-positive T cells (Figure 7). This anti-inflammatory effect translated into a drastic reduction in the number of glomeruli affected by crescents and the increase in normally appearing glomeruli accompanied by very low levels of proteinuria. These data suggest that cAng-(1-7) administered on top of standard care could be a novel clinical approach for ANCA-GN, particularly for those patients who do not respond completely to conventional treatment, and could also be a strategy for minimizing the use of immunosuppressants and avoiding their severe side effects.

## Figures and Tables

**Figure 1 cells-11-02434-f001:**
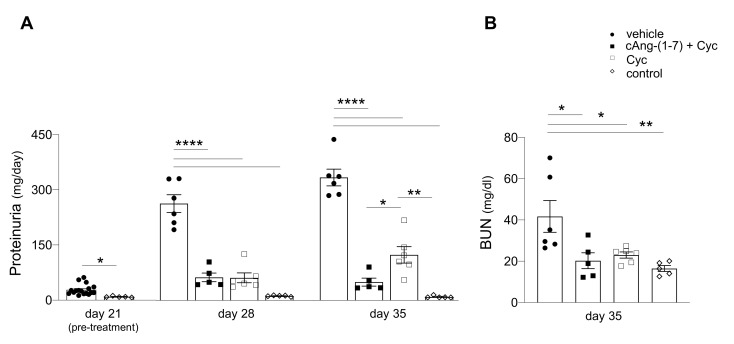
Treatment with cyclic Angiotensin-(1-7) (cAng-(1-7)) on top of cyclophosphamide (Cyc) blocks the rise of proteinuria and the impairment of renal function. (**A**) Time course of urinary protein excretion measured in WKY rats with ANCA-GN treated with vehicle (*n* = 6), cAng-(1-7) plus cyclophosphamide (*n* = 5) and cyclophosphamide alone (*n* = 6), and WKY controls (*n* = 5). Data are mean ± SEM. (**B**) Renal function evaluated as blood urea nitrogen (BUN) levels measured at the end of the study (day 35). Data are mean ± SEM. * *p* < 0.05, ** *p* < 0.01, **** *p* < 0.0001.

**Figure 2 cells-11-02434-f002:**
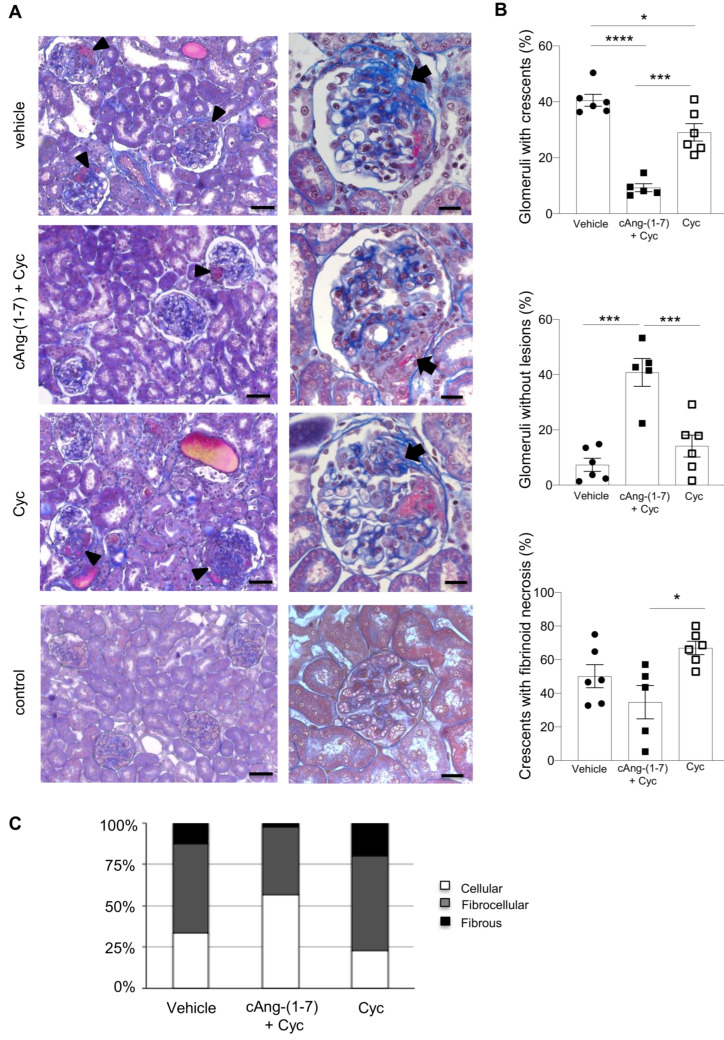
Addition of c-Ang-(1-7) to cyclophosphamide preserves kidney structure. (**A**) Representative images of Masson’s trichrome-stained kidneys from WKY rats with ANCA-GN treated with vehicle, cAng-(1-7) plus cyclophosphamide, or cyclophosphamide alone, and WKY control rats at day 35; left column (×200 Magnification; scale bar 50 µm): crescents are indicated by arrowheads; right column (×630 Magnification; scale bar 20 µm): arrows indicate crescentic lesions. (**B**) Glomerular crescents and normal-appearing glomeruli are expressed as percentage of total glomeruli. Fibrinoid necrosis is expressed as percentage of glomeruli with crescentic lesions. Data are mean ± SEM. * *p* < 0.05, *** *p* < 0.001, **** *p* < 0.0001. Closed circle, vehicle; closed square, cAng-(1-7) + Cyc; open square, cyclophosphamide. (**C**) Prevalence of cellular, fibro-cellular and fibrous crescents in each group of rats with ANCA-GN. Data are expressed as mean percentage of glomerular crescents.

**Figure 3 cells-11-02434-f003:**
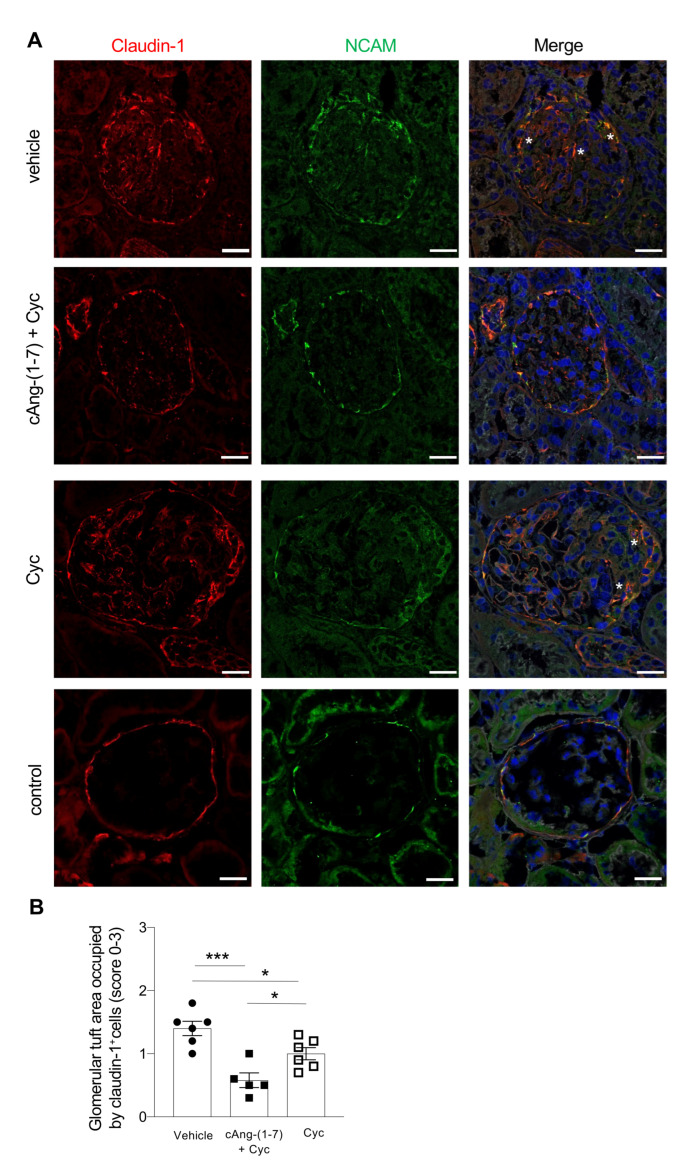
cAng-(1-7) plus cyclophosphamide re-established the normal distribution of PECs/progenitor cells. (**A**) PECs with a progenitor cell phenotype are identified by double staining for claudin-1 (red) and NCAM (green). Nuclei are stained with DAPI (blue) and renal structure with lectin (white). Magnification: ×630; scale bar 25 μm. (**B**) Altered distribution of PECs in the glomeruli is expressed with a score from 0 to 3 related to the percentage of the glomerular tuft area occupied by claudin-1-positive cells. Data are mean ± SEM. * *p* < 0.05, *** *p* < 0.001. Closed circle, vehicle; closed square, cAng-(1-7) + Cyc; open square, cyclophosphamide.

**Figure 4 cells-11-02434-f004:**
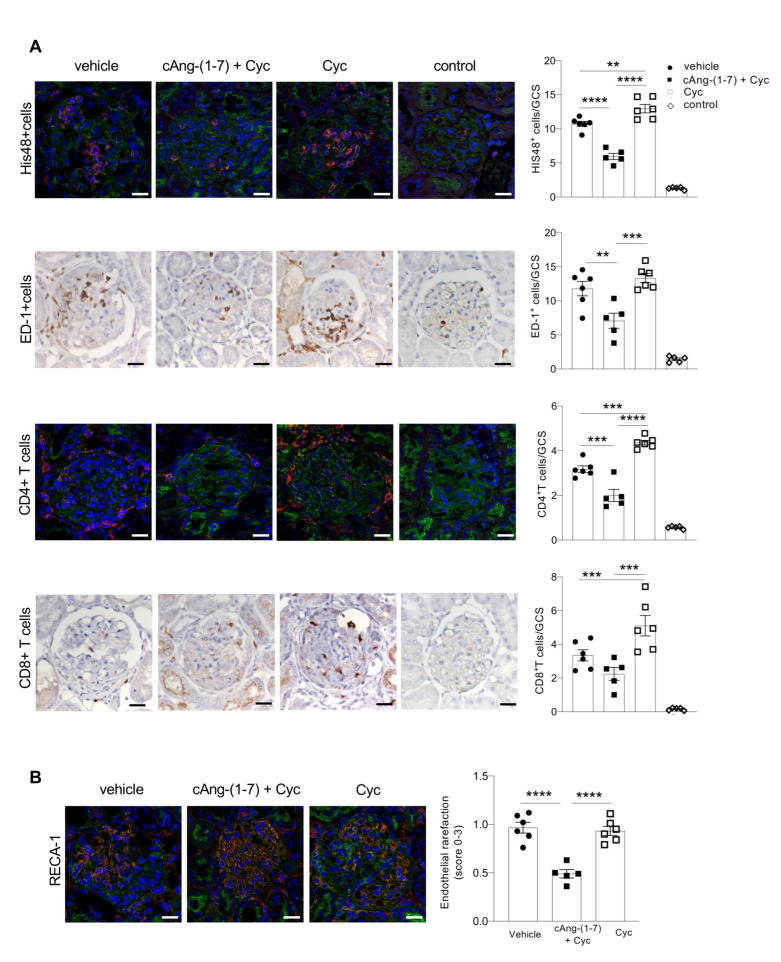
Addition of cAng-(1-7) to cyclophosphamide limits glomerular inflammation and endothelial rarefaction in rats with ANCA-GN. (**A**) Representative images and quantification of inflammatory/immune cells expressed as an average number of cells per glomerulus showing, respectively, HIS48^+^ granulocytes (red), ED1^+^ monocytes/macrophages (brown), CD4^+^ T cells (red), and CD8^+^ T cells (brown) in the glomeruli of WKY rats with ANCA-GN treated with vehicle, cAng-(1-7) plus cyclophosphamide or cyclophosphamide alone and WKY controls. Magnification: ×630; scale bar 25 µm. (**B**) Representative images showing glomerular RECA-1 expression in WKY rats with ANCA-GN treated with vehicle, combined therapy, and cyclophosphamide alone. The distribution of RECA-1 staining of capillaries in the glomerular tuft was scored as follows: 0, normal density of capillaries; 1+, reduced capillary density in <25% of tuft; 2+, reduced capillary density in 25%–50% of the tuft; and 3+, reduced capillary density in >50% of the tuft. Data are mean ± SEM. ** *p* < 0.01, *** *p* < 0.001, **** *p* < 0.0001. Magnification: ×630; scale bar 25 µm. Closed circle, vehicle; closed square, cAng-(1-7) + Cyc; open square, cyclophosphamide.

**Figure 5 cells-11-02434-f005:**
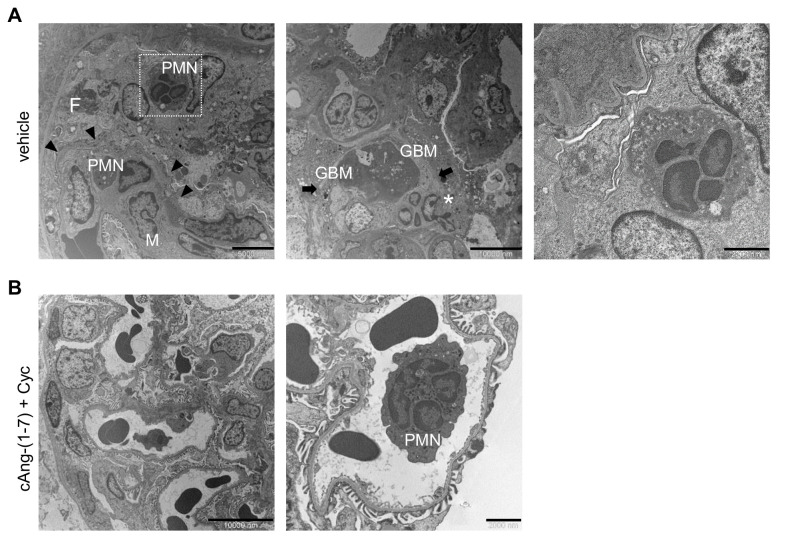
Addition of cAng-(1-7) to cyclophosphamide ameliorates glomerular ultrastructural abnormalities in glomeruli of rats with ANCA-GN. (**A**) Representative transmission electron microscopy images of glomeruli in WKY rats with ANCA-GN given vehicle showing the following: left panel, a crescentic lesion with fibrin deposits (F) associated with accumulation of polymorphonuclear neutrophils (PMN) and mononuclear cells (M). Arrowheads indicate podocyte foot process effacement; middle panel, glomerular basement membrane (GBM) ruptures (arrows), and cellular extravasation (asterisk); right panel, insert of the left panel showing a degranulated neutrophil. (**B**) Representative glomeruli from rats with ANCA-GN treated with cAng-(1-7) plus cyclophosphamide showing the following: left panel, well-preserved glomerular architecture; right panel, circulating non-degranulated neutrophil (PMN). Scale bars: (**A**) 5000, 10,000, 2000 nm; (**B**) 1000, 2000 nm.

**Figure 6 cells-11-02434-f006:**
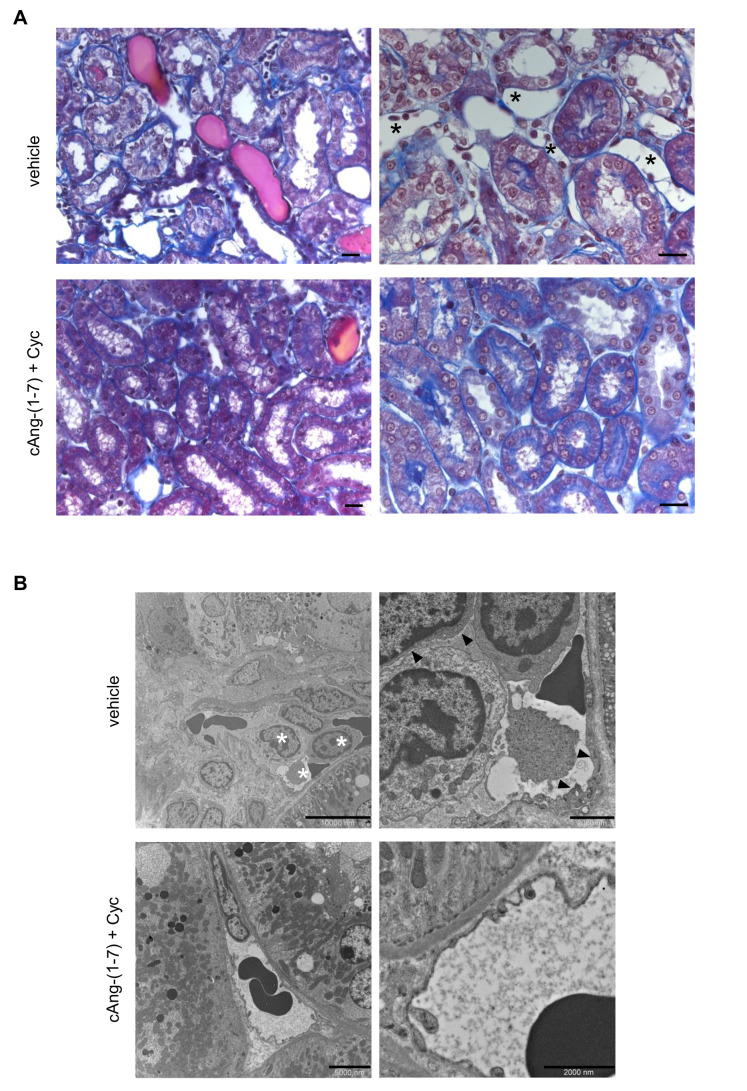
Addition of cAng-(1-7) to cyclophosphamide ameliorates tubulo-interstitial abnormalities in rats with ANCA-GN. (**A**) Representative images of Masson’s trichrome-stained kidney sections from WKY rats with ANCA-GN treated with vehicle and cAng-(1-7) plus cyclophosphamide: left panels, tubulointerstitial alterations consisting of tubular casts, tubular dilatation and atrophy in vehicle-treated group limited by the combined therapy (×400 Magnification; scale bar 20 µm); right panels, dilated peritubular capillaries (asterisks) with inflammatory cells (capillaritis) in vehicle-treated group prevented in rats that received cAng-(1-7) plus cyclophosphamide (×630 Magnification; scale bar 20 µm). (**B**) Representative micrographs from transmission electron microscopy of interstitial area in vehicle group and cAng-(1-7) plus cyclophosphamide group: left panel, accumulation of inflammatory cells (asterisk) in peritubular capillaries of vehicle-treated group abolished by the combined therapy; right panel, swelling of endothelial cells and loss of fenestrations (arrowheads) in WKY rats given vehicle and preserved peritubular endothelial compartment in WKY rats treated with the combined therapy. Scale bars: 10,000, 2000, 5000, 2000 nm.

**Figure 7 cells-11-02434-f007:**
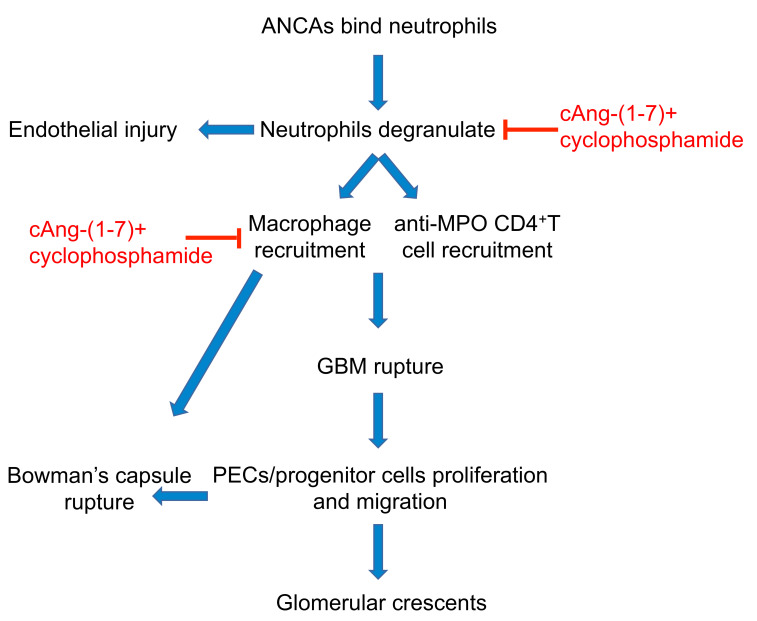
Schematic representation of the sequence of events that cause crescentic lesion formation in WKY rats with MPO-ANCA-glomerulonephritis, and possible targets of cAng-(1-7) + cyclophosphamide therapy. ANCA, anti-neutrophil cytoplasmic antibodies; MPO, myeloperoxidase; cAng-(1-7), cyclic angiotensin 1-7; GBM, glomerular basement membrane; PECs, parietal epithelial cells.

**Table 1 cells-11-02434-t001:** Systemic Parameters in Rats with ANCA-GN.

Groups	Body Weight (g)	SBP (mmHg)	Kidney Weight/ Body Weight (%)
Vehicle (*n* = 6)	272 ± 7 **	192 ± 7 ***	0.69 ± 0.05 ***
cAng-(1-7) + Cyc (*n* = 5)	301 ± 7	168 ± 3	0.46 ± 0.01
Cyclophosphamide (*n* = 6)	289 ± 8	182 ± 5 **	0.54 ± 0.03 *
Control (*n* = 5)	326 ± 15	148 ± 3	0.37 ± 0.01

Data are mean ± SEM. * *p* < 0.05; ** *p* < 0.01; *** *p* < 0.001 versus control. *p* < 0.05; *p* < 0.001 vs. vehicle. cAng-(1-7): cyclic Angiotensin-(1-7); Cyc: cyclophosphamide; SBP: systolic blood pressure.

## Data Availability

Data presented in this manuscript are available from the corresponding author on reasonable request.

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
