# Peer review of "Add-On Cyclic Angiotensin-(1-7) with Cyclophosphamide Arrests Progressive Kidney Disease in Rats with ANCA Associated Glomerulonephritis"

_cells, 2022, doi:10.3390/cells11152434_

Round 1
Reviewer 1 Report
In a rat model of crescentic glomerulonephritis associated with anti-neutrophil cytoplasmic antibodies, the combined application of cyclic angiotensin-(1-7) with cyclophosphamide enhanced markedly the kidney-protective effect of cyclophosphamide alone. The manuscript is well written and the conclusions are sound.
My remarks concern two items in the discussion.
In discussion, the role of angiotensin II in favoring destructive processes should be better described. At inflammatory sites, neutrophil-derived serine proteases are known to release angiotensin II from angiotensin I and angiotensinogen. The corresponding references should be included. Some words about effects of ACE2 on angiotensin II would be also helpful.
A scheme would be useful to present the anti-inflammatory effects of cyclic angiotensin (1-7) and cyclophosphamide.
Author Response
In a rat model of crescentic glomerulonephritis associated with anti-neutrophil cytoplasmic antibodies, the combined application of cyclic angiotensin-(1-7) with cyclophosphamide enhanced markedly the kidney-protective effect of cyclophosphamide alone. The manuscript is well written and the conclusions are sound.
My remarks concern two items in the discussion.
In discussion, the role of angiotensin II in favoring destructive processes should be better described. At inflammatory sites, neutrophil-derived serine proteases are known to release angiotensin II from angiotensin I and angiotensinogen. The corresponding references should be included. Some words about effects of ACE2 on angiotensin II would be also helpful.
Re: We are very grateful to the reviewer for the appreciation of our work. As suggested, new statements have been added in the discussion (page 16) and new references have been quoted accordingly (ref. 32-34). A statement on ACE2 has been included in the Introduction (page 2).
A scheme would be useful to present the anti-inflammatory effects of cyclic angiotensin (1-7) and cyclophosphamide.
Re: A scheme has been included in the revised version according to the reviewer’s recommendation (new Figure 7, page 17).
Reviewer 2 Report
In this paper, Cerullo et al. used a rat model of ANCA-associated vasculitis to show that adding cANG-(1-7) to a partially effective dose of cyclophosphamide (Cyc) arrests renal disease progression, supporting the concept that this treatment could be tested as an immunosuppressant sparing approach.
This is an interesting topic from an established group. However, enthusiasm is limited by the following concerns:
1. Figure 1: the benefit of cANG-(1-7) + Cyc over Cyc treatment alone is marginal and limited to proteinuria at 35 days. This is matter of concern. A control group with cANG-(1-7) alone should be included to dissect the contribution of this therapy alone.
2. In figure 2, the effects of Cyc alone on histological lesions seems negligible compared to vehicle. How do the authors reconcile this finding with the significant differences in BUN between vehicle and Cyc? How many pathologists did read the kidney sections? Were they blind to animal treatment group?
3. Figure 3 lacks data quantification.
4. Figure 4: It is concerning that glomerular granulocytes and T cells were more frequent in Cyc- compared to vehicle-treated animals. What explanation do the authors provide for this counterintuitive finding?
Minor
1. Table 1 should include the number of rats per group.
2. It is unclear what was the sex of the rats used for the experiments. Due to different susceptibility to disease between sexes, the authors should provide this information. Ideally, they should perform some of their experiments in both sexes.
Author Response
Comments and Suggestions for Authors
In this paper, Cerullo et al. used a rat model of ANCA-associated vasculitis to show that adding cANG-(1-7) to a partially effective dose of cyclophosphamide (Cyc) arrests renal disease progression, supporting the concept that this treatment could be tested as an immunosuppressant sparing approach.
This is an interesting topic from an established group. However, enthusiasm is limited by the following concerns:
- Figure 1: the benefit of cANG-(1-7) + Cyc over Cyc treatment alone is marginal and limited to proteinuria at 35 days. This is matter of concern. A control group with cANG-(1-7) alone should be included to dissect the contribution of this therapy alone.
Re: The reviewer is correct. Indeed, at day 28 cAng-(1-7) + cyclophosphamide and cyclophosphamide alone had a similar effect on proteinuria, while later on the combined treatment maintained a stable and more pronounced antiproteinuric effect than cyclophosphamide. In the discussion we underlined that a rise in proteinuria at the end of the study in rats treated with cyclophosphamide alone indicates a tendency for the disease to relapse in the absence of co-administered cAng(-1-7). Of note, two-way ANOVA analysis for repeated measurements of proteinuria revealed that the progression of proteinuria observed in the group of animals treated with cyclophosphamide alone, was of statistical significance (p<0.05) indicating progression of the disease with time. On the contrary, in the combined therapy group the same analysis revealed that proteinuria levels at the end of the study were comparable to levels measured at day 21 (pre-treatment) indicating that the treatment with cAng-(1-7) + cyclophosphamide arrests the progression of proteinuria. Notably, at day 35, the combined therapy accounted for a reduction of proteinuria of 60% with respect to cyclophosphamide alone.
We understand the reviewer’s point that a control group with cAng-(1-7) could dissect the contribution of each treatment alone, but our study was designed to better accomplish the translatability of the experimental data to the clinics where the use of cAng-(1-7) alone could not be feasible. Our goal was to mimic the clinical condition of ANCA-GN patients who may benefit from the addition to the standard therapy of a novel molecule, which could allow minimizing the use of immunosuppressants and their toxicity.
- In figure 2, the effects of Cyc alone on histological lesions seems negligible compared to vehicle. How do the authors reconcile this finding with the significant differences in BUN between vehicle and Cyc? How many pathologists did read the kidney sections? Were they blind to animal treatment group?
Re: Reduction of histological lesions is a hard end point, and the effect of cyclophosphamide of reducing glomerular crescents by 30% compared with vehicle (p<0.05) was enough to restore BUN to control levels and to reduce proteinuria. Kidney sections have been analyzed by two expert pathologists blind to treatment groups, who were in agreement with the results. A statement has been added in ‘Materials and Methods’ section, page 3.
- Figure 3 lacks data quantification.
Re: As requested by the reviewer, we provide in the revised version data quantification on the altered distribution of PECs in the glomeruli of ANCA-GN rats and the effect of the treatments (Figure 3B).
- Figure 4: It is concerning that glomerular granulocytes and T cells were more frequent in Cyc- compared to vehicle-treated animals. What explanation do the authors provide for this counterintuitive finding?
Re: We have no explanation for this interesting point. In our defence, we can say the following: first of all this model has a high variability in inflammatory response; second, we cannot exclude that after a single injection of cyclophosphamide that is not enough to fully control the disease progression, a rebound phenomenon may occur by which inflammatory cells tend to be more represented when the effect of the drug is waining.
Minor
- Table 1 should include the number of rats per group.
Re: The numerosity of the groups has been added in the table.
- It is unclear what was the sex of the rats used for the experiments. Due to different susceptibility to disease between sexes, the authors should provide this information. Ideally, they should perform some of their experiments in both sexes.
Re: In this study we used male rats as it was reported in the “Animals” section of Materials and Methods of the original version (line 75).
Round 2
Reviewer 2 Report
The authors adequately addressed all my concerns.